# Outdoor Thermal Comfort Integrated with Energy Consumption for Urban Block Design Optimization: A Study of the Hot-Summer Mediterranean City of Irbid, Jordan

**Mohammad Mazen Khraiwesh [1] and Paolo Vincenzo Genovese [2],***

1 School of Architecture, Tianjin University, Weijin Road Campus, Tianjin 300072, China; sawaie2009@gmail.com
2 College of Civil Engineering and Architecture, Zhejiang University, Hangzhou 310058, China
* Correspondence: pavic@zju.edu.cn

**Abstract:** With an increasing awareness of urban health and well-being, this study highlights the growing importance of considering environmental quality in urban design beyond mere energy performance. This study integrates outdoor and indoor quality by investigating the effect of design parameters at an urban block scale (building form restricted to width and length as rectangular and square, building orientation, block orientation, building combination, building height, facade length, built-up percentage, setbacks, and canyon aspect ratio) on outdoor thermal comfort and energy use intensity. In addition, it explains the different correlations between outdoor thermal comfort and energy use intensity in different urban block designs in a hot-summer Mediterranean climate in Jordan. The study adopts a performance-driven approach using simulation tools of Ladybug, Honeybee, Dragonfly, and Eddy3d plugins across the grasshopper interface and evaluates 59 different urban block designs with nine different orientations (0°, 1°, 45°, 85°, 87°, 90°, 355°, 358°, and 359°). The results show that there is a positive correlation between the canyon aspect ratio and the environmental performance of the urban block designs. North–south street canyons are more effective at enhancing microclimates. Negatively increasing the street aspect ratio by more than four affected outdoor thermal comfort by increasing longwave radiation. Further results suggest a positive correlation between the compactness of urban blocks and their environmental performance, with north–south street canyons found to be more effective in enhancing microclimates. The study emphasizes the need to understand the distribution of open spaces formed by buildings and to strike a balance between day and night, as well as summer and winter conditions in outdoor spaces.

**Keywords:** performance-driven urban design; outdoor thermal comfort; universal thermal climate index (UTCI); hot-summer Mediterranean climate; energy consumption; grasshopper

## 1. Introduction

The current global population has more than tripled since the mid-twentieth century. This population explosion is estimated to reach approximately 8.5 billion by 2030 and increase by 1.18 billion in the following two decades, reaching 9.7 billion in 2050 [1]. Jordan is currently experiencing tremendous population growth due to the influx of refugees from neighboring countries and an increase in the birth rate among the native Jordanian population. This has led to the rapid urbanization of some cities to meet housing and infrastructure needs [2], combined with random urban sprawl with structures that are poorly designed environmentally [3]. The growth of cities is associated with severe environmental consequences, thus affecting the quality of life of their inhabitants [4]. It is possible for residential energy consumption and structures to change significantly as a result of urbanization [5], whereas urbanization is the process of transferring population from rural to urban areas, which can increase energy demand through various channels,

including the growth of urban populations and the extension of construction [6]. As energy consumption has increased, $CO_2$ emissions have increased [7], and this has been considered a major contributing factor with regard to climate change [8]. Furthermore, urban densification is often characterized by narrow canyons with impervious construction materials, reduced vegetation, and increased pollution, causing more sensible heat storage, shifting longwave heat emissions, entrapping shortwave radiation within street level, and hindering evaporative and convective cooling, affecting an urban heat island (UHI) [9].

According to the definition of outdoor thermal comfort (OTC), it is the feeling of satisfaction with a particular thermal environment [10]. A cooler urban temperature could reduce the cooling load on buildings, resulting in energy savings [11]. Furthermore, as people spend more time in outdoor spaces, it will reduce heating and cooling energy usage and the use of other electronic devices [12]. A key factor in determining the quality of outdoor spaces is the outdoor microclimate. Pedestrians are directly exposed to their immediate surroundings in terms of the sun and shade, wind speed, and other elements, as opposed to commuters. Therefore, a microclimate greatly influences people's perception of thermal comfort [13]. In his study, "Life Between Buildings: Using Public Space", Gehl [14] demonstrated, through counting people sitting on sunny and shady benches, the effect of microclimates on outdoor activities.

Since the urban microclimate is determined by the spatial structure of a city, the structure of blocks, building shapes, open space design, etc. [15], the urban microclimate affects the cooling and heating loads of buildings as well as the outdoor thermal comfort, and thus the performance of buildings. It is, therefore, one of the most important factors to be considered in urban design [16]. With the increasing awareness of urban health and well-being, urban environmental analysis should expand from energy performance to new considerations based on environmental quality. The limited ability to annually assess outdoor thermal comfort, which is the primary focus of these considerations, has limited research into the relationships between urban morphology and annual energy performance [17]. The scope of codes and standards has been limited to maintaining the thermal neutrality of indoor spaces and has neglected the thermal impact on the surrounding environments [18], while the interactions between the exterior envelope of buildings and the outdoor thermal field perceived by pedestrians have been recently pointed out [19]. The importance of this study is combining enhanced building energy use with OTC to find the balance between indoor and outdoor environments without neglecting either of them.

The urban block, as one of the important morphological elements, took on new dimensions at the beginning of the 20th century, when reforming urban blocks became a global trend [20]. Planners and architects understood that focusing only on individual buildings was inadequate, and that it was important to also consider groups of buildings or entire urban blocks [21]. The size and shape of urban blocks effectively contribute to the formation of the character of the environment [22]. Moreover, the block sides respond to the internal and external loads of the blocks, as well as the buildings and the street structure, respectively [23]. It is much easier to implement strategic adjustments at the urban block level [24]. However, concerns have been raised about the lack of studies related to urban environmental analysis at the urban block scale [25,26]. Numerous studies have focused on investigating the energy consumption of individual buildings; however, assessing this on a larger scale requires further exploration. When integrating indoor and outdoor thermal parameters, it is essential to recognize how each building within an urban block has an influence on the others, ultimately giving a fuller picture of the energy performance of the overall area. Unfortunately, building regulations tend to disregard the role of urban morphology and architecture when developing sustainable cities, oftentimes resulting in uniform-height buildings being a dominant feature [27].

In the last few years, urban areas in Jordan have been formulated by a combination of urban blocks based on a set of regulations [28], accompanied by rapid urbanization and random sprawl [29]. These regulations have been implemented without any numerical or

analytical studies that consider urbanization, regardless of whether data to enhance these urban areas responds to the local climate conditions. The climate in Jordan is predominantly Mediterranean, with hot, dry summers and wet, cool winters [30]. Over 60% of the energy consumed in the residential sector was used for space heating and cooling in 2020 [31]. Furthermore, Jordan's moving from one summer peak pattern to two flood peaks (winter peak pattern and summer peak pattern) has been noted in Jordan yearly, with the peaks themselves increasing annually due to the increase in demand. Studies investigating the correlation between OTC, EUI, and urban form design mostly preferred to exclude containing more than one climate condition due to the complexity of such multi-aspect methods, while different climate conditions should be considered.

In summary, the main contributions of this paper are as follows:

- This study integrates outdoor and indoor quality by investigating the effect of design parameters at an urban block scale (building form restricted to width and length as rectangular and square, building orientation, block orientation, building combination, building height, facade length, built-up percentage (BUP), setbacks, and CAR on OTC and EUI). In addition, it explains the different correlations between OTC and EUI in different scenarios of urban block design.
- A detailed numerical and physical analysis of the effects of urban block design parameters on EUI and OTC as well as the effects of different climatic factors with different weather conditions (annual, summer, and winter) and different spatial zones; therefore, this research includes a comparative analysis of 59 residential urban block designs with three different residential types (type A, type B, and type C) in Irbid, north Jordan, in a hot-summer Mediterranean climate.
- The computational workflow adopted performance-driven urban morphology using simulation tools (Grasshopper interface plugins; Eddy3d, Ladybug, Honeybee energy, and UTCI) following the guidelines of the latest studies and included a modeling approach, which enables estimating the effects of the urban heat island effect on a microclimate to use the validated data for estimating the OTC and EUI of the urban block design. This approach using Grasshopper can be deemed appropriate to visualize detailed results that could be helpful for decision-makers and urban designers at early design stages.
- Based on the achievements relying on the correlations between these different parameters and OTC and EUI, this paper defines the acceptable range of design parameters to maximize outdoor thermal comfort while reducing energy consumption at the same time, in addition to proposing an optimized urban block design.
- By using computational numerical simulations, this study presents design strategies and guidelines to assist designers and decision-makers in optimizing and improving urban communities through increasing both energy efficiency and satisfaction in outdoor environments.

## 2. Related Work

In the related context of OTC and building energy consumption, several studies investigated the effect of urban design parameters on these factors. In this section, we mainly summarize the work related to the effect of different urban design parameters and their impact on EUI and OTC.

Ibrahim et al. (2021) investigated the impact of changing the morphological characteristics of three-block typologies in the Mediterranean city of Cairo, Egypt and their related parameters to understand their multidimensional relationship with environmental conditions, outdoor thermal comfort and energy use intensity (EUI) [32]. Abdallah (2015) studied the influence of open spaces between buildings with a building height-to-street width ratio or canyon aspect ratio (CAR) (H/W) of 0.24~0.6 in one of the urban patterns of new housing sectors in New Assiut city and of deep canyons with H/W ratio of 4 in one of the new residential houses of El-Abrahimia and El-Moalemen complexes in the center of Assiut city in Egypt on indoor thermal comfort (energy consumption), and he concluded

in his study that the effect on indoor thermal comfort is an index of the influence of deep canyons with H/W ratio [33]. The relationship between sky-view factor (SVF) and the effect of urban heat island (UHI) in Montreal has been investigated in the study of Wang and Akbari by evaluating the effects of SVF on air temperature ($T_a$) and mean radiant temperature (MRT). They concluded that the amount of energy used for indoor heating and air conditioning is affected by $T_a$, while the value of MRT is the sum of all shortwave and longwave radiation fluxes that are absorbed by the human body and affect human energy balance and thermal comfort [34].

Computational numerical simulation has several advantages, and its useful applications have been proven by several studies. Kamel (2021) presented a comprehensive simulation workflow of the built environment using the Ladybug, Honeybee and Butterfly plugins in the Grasshopper interface to create two different metrics in order to measure the outdoor thermal comfort for pedestrians in urban street canyons, MRT and the universal thermal climate index (UTCI) based on the solar reflective index (SRI), where a relationship between urban microclimate, building energy use and outdoor thermal comfort was found by studying two urban neighborhoods in Egypt [35]. Hamdan and Oliveira (2019) investigated the impact of urban design strategies on a microclimate. The study conducted an investigation into canyon ratio, orientation, vegetation shading and wind speed using the case study of Al Ain city in the UAE by performing simulations using Grasshopper with OpenStudio, EnergyPlus and Radiance plugins, and UTCI [36]. On the other hand, few studies have adopted computational simulation methods in determining the correlation between OTC, energy use intensity (EUI) and urban form. Natanian et al. (2020) carried out a study that filled the gap between urban morphology. OTC combined with EUI was found to be insufficient due to the incapability of annual outdoor thermal comfort evaluation and the limitation of the exploration of the interrelationships between urban morphology and annual energy performance by utilizing the capabilities of Eddy3d—a Grasshopper plugin that provides effective calculations of hourly microclimatic wind factors via Open-FOAM, which are then used to create annual outdoor thermal comfort plots. They utilized this method to conduct a parametric study in three different hot climates for five different typologies in different density scenarios. A total of 60 design iterations for energy performance, OTC, and self-shading by shade index have been evaluated. They found that there is a high correlation between the annual shade index and OTC in all climatic contexts, which indicates the potential of the shade index as an effective indicator. In addition, they noted a superiority of one typology in comparison with other different typologies in both EUI and OTCA studies [17]. Mirzabeigi and Razkenari (2022) proposed an optimization framework using Grasshopper plugins, Ladybug tools and Eddy3d to compare six varying urban typologies of low-rise, mid-rise and high-rise categories based on OTC and EUI performance and identify a set of the best design solutions in the conceptual urban typologies in Syracuse, United States [37].

Despite the above-mentioned studies, there are a few that have noted the correlations between outdoor thermal comfort (OTC) and energy use affected by the parameters of building geometry and urban design, which create urban performance. This is due to the tremendous spatial and temporal variations in microclimatic conditions surrounding pedestrians, e.g., air temperature, wind speed, humidity and solar radiation, and the interaction between these conditions and the way pedestrians respond physically and physiologically to achieve psychological satisfaction. Furthermore, winter days have been neglected, whether in OTC studies or EUI studies, where most commonly thermal perception studies have been conducted in relatively narrow condition ranges [38], such as summertime [39,40], warm seasons [41] or climatic zones with small annual air temperature amplitudes (i.e., tropical and subtropical climates). When considering temperate climates and their general impact on solar radiation in Mediterranean cities, one must also consider that there is an increased need for solar radiation in the winter when the temperature is lower than 10 °C [42]. The detailed abbreviations and definitions used in the paper are listed in Abbreviations section.

## 3. Materials and Methods

This study adopted a performance-driven approach using simulation tools (Grasshopper interface plugins; Eddy3d, Ladybug, Dragonfly, Honeybee energy and UTCI) with validated simulation engines (EnergyPlus and OpenFOAM), following the guidelines of the latest studies [18,32,43–50]. It included a modeling approach, which enables estimating the effects of the urban heat island effect on a microclimate using the validated data for estimating the OTC and EUI of the urban block design. This approach has been used on 59 different urban block designs in Irbid, Jordan.

### 3.1. Study Area

Jordan is a country in the Middle East, an area known as the Mediterranean. The country is located in the western part of Asia, with latitudes of 31.96° N and longitudes of 35.93° E, and has a total area of 89,320 km$^2$. According to statistical data, in 2021, Jordan had a population of approximately 10,269,022 people. The country's population density is 115 per km$^2$, and 91.5% of the population is urban. The country has 12 governorates, among which are Amman and Irbid. Amman is the capital city of Jordan, where 42% of the country's population lives. Irbid governorate in northern Jordan has a total area of 1571.8 km$^2$. Irbid governorate is the second most populated governorate after the Amman governorate, with a population of around 2,003,800, and the highest population density in the country at 1126.2 per km$^2$. Irbid city is the capital of the Irbid governorate, with a total area of 30 km$^2$. Irbid (32.5568° N, 35.8469° E) is the second-largest metropolitan population in Jordan after Amman, with a population of around 569,068. Irbid has a hot-summer Mediterranean climate (Köppen: Csa) [51]. The city's yearly temperature is 21.93 °C, and it is −0.24% lower than Jordan's averages. Irbid typically receives about 19.55 mm of precipitation and has 44.18 rainy days annually (12.1% of the year).

### 3.2. Site Selection, Scenarios and Time Periods

Irbid, as the capital of Irbid governorate, has been chosen as a location where urban block designs will be applied to our study. Irbid is located in the north of Jordan, about 70 km north of the capital of Amman and approximately 20 km south of the Syrian border. By understanding the urban development of Irbid, we decided to choose 6 different urban block designs; two from each of the different residential regulation types (residential type A, residential type B, and residential type C) (see Section 3.4) are concentrated in the south-east of Irbid, where the local government decided to establish a new residential zone divided into urban blocks in the last few years. The coordinates of the sites are as follows: the first site (32.519780° N, 35.837574° E), the second site (32.521113° N, 35.837034° E), the third site (32.531585° N, 35.828420° E), the fourth site (32.518358° N, 35.903247° E), the fifth site (32.517213° N, 35.907899° E), and the sixth site (32.522726° N, 35.832724° E). These selected sites are highlighted in Figure 1.

Figure 2 shows urban block design scenarios based on urban block designs that have been selected for the study and to avoid deformation due to the streets, where the exact parameters, parcels and features were derived by using a shape file from Greater Irbid Municipality and validated with the data from the Department of Land and Survey and attached [52]. While the study for analyzing EUI (cooling, heating, and total energy) and OUT will be conducted in three periods: the annual period from 1 January to 31 December, the summer season from 25 May to 8 October, and the winter season from 5 December to 10 March.

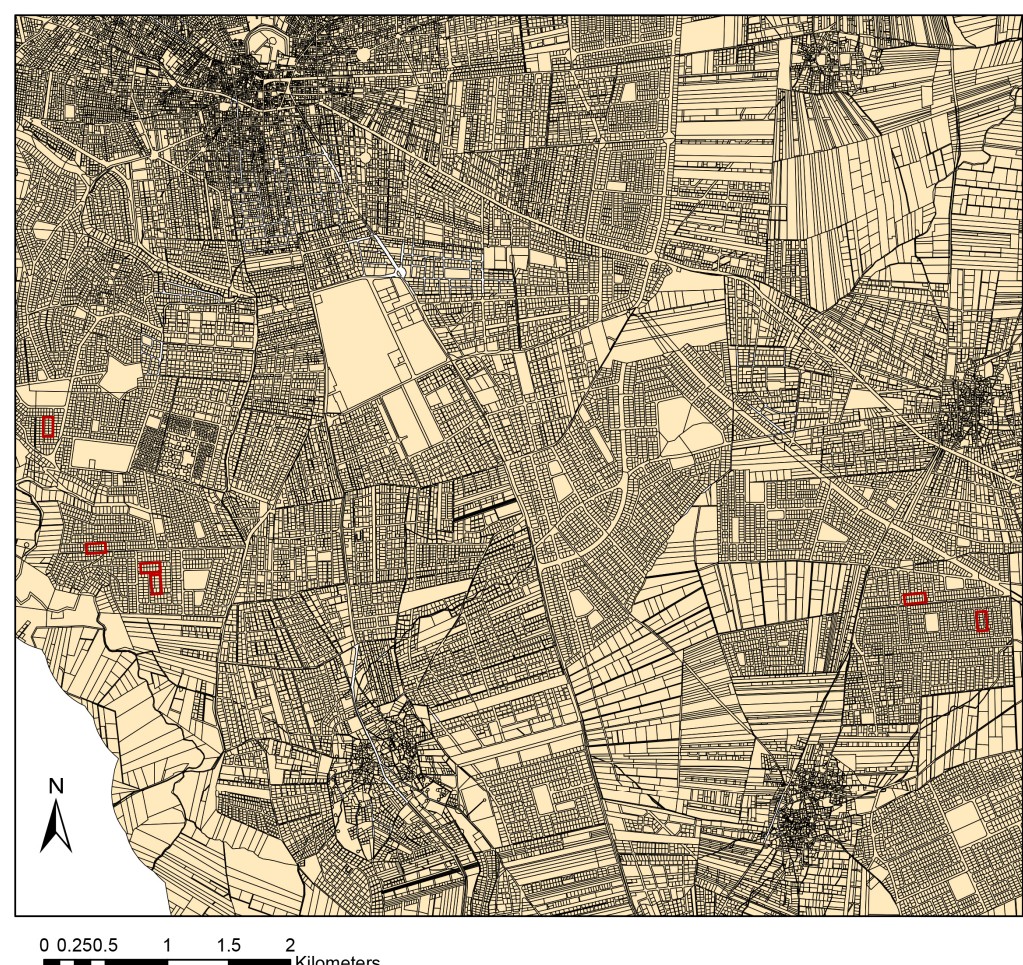

**Figure 1.** The location of the selected sites within the map of Irbid.

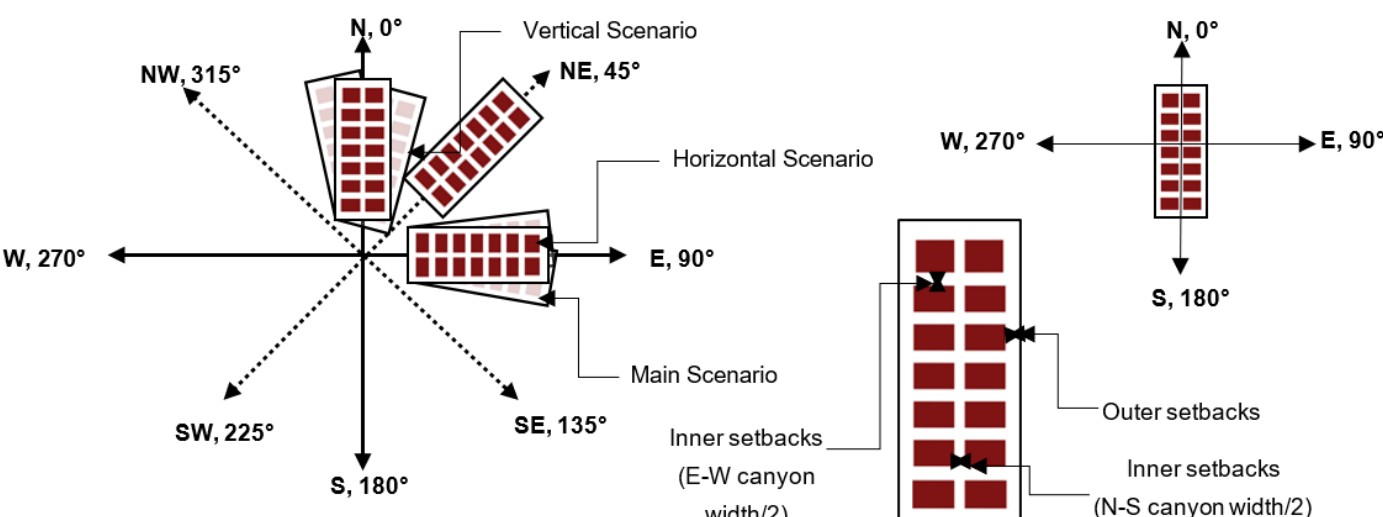

**Figure 2.** Urban block scenario scheme in the study.

### 3.3. Analytical Workflow

Figure 3 presents the analytical workflow diagram performed and tested in this study, which clarifies data streaming between various engines and demonstrates the relationship between different resources from the inputs to the outputs. The analytical workflow is

divided into three stages or phases: inputs (see Section 3.4), weather data morphing (see Section 3.5) and output visualization (see Section 3.6).

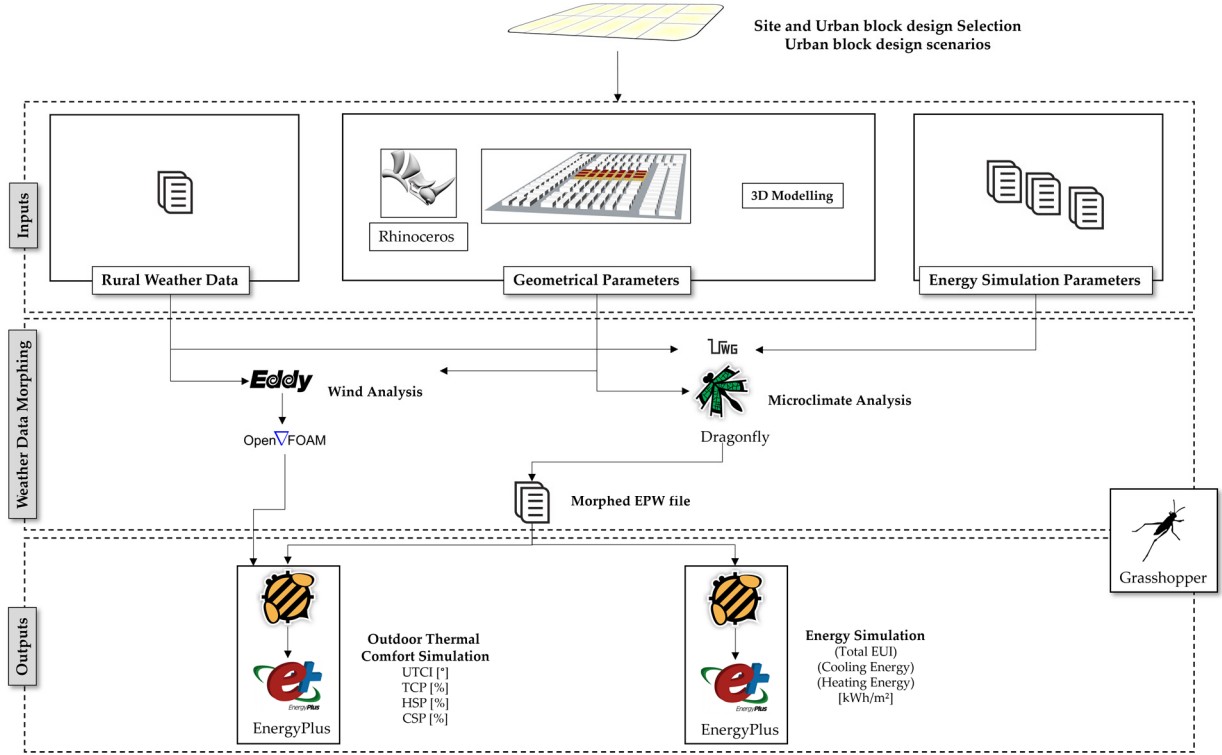

**Figure 3.** Analytical workflow in Grasshopper shows the data flow between input and output components in Grasshopper.

Each of these phases or stages includes the correlated parts, explained and illustrated in detail in the corresponding sections. Once the urban block sites are selected, the next step is the first phase of the analytical workflow, i.e., the input phase. The input phase considers input parameters and data and has three parts, geometrical parameters (3D modeling) (see Section 3.4), rural weather data and simulation parameters, where the latter two are explained under the urban weather generator (UWG) section. The next stage is weather data morphing (see Section 3.5) and has two parts: UWG to estimate the hourly urban canopy air temperature and relative humidity using data streamed from rural weather data in the first phase and wind speed calculation (see Section 3.5.2). The third phase is the performance or output visualization and includes two parts: energy use simulation (see Section 3.6.1) and OTC simulation (see Section 3.6.2).

### 3.4. 3D Modeling

After selecting the sites, the second step is geometrical modeling using Grasshopper to determine the current situation of the selected sites' empty lands or semi-empty lands; therefore, the urban block modeling was based on three steps as follows:

- Creating the urban block footprints: The selected sites have been divided into parcels that have been defined by the local authorities (Greater Irbid Municipality and Department of Land and Survey) and each parcel has a specific land-use, whereas our study only focuses on the residential sector. In addition, each residential land-use has a different residential type with different regulations according to Irbid's residential building regulations [53]. Table 1 shows the selected residential types with their correlated rules and restrictions. These rules define the urban block parameters, while the building forms and building and urban block orientations have been neglected by the local regulations as well as the constant building heights. Our study will cover

these parameters and has been created to be appropriate with these regulations. In terms of building forms, the study will include only square and rectangular forms, according to Ali and Hikmat's study, which found that the geometrical layout of the residential buildings in Jordan is divided into 61% rectangular and 27.2% square, with a few percentages with a U-shape or an L-shape [54].

- Urban block modeling: This step creates the urban block models as an extrusion from the urban block footprints that have been defined in the previous step. In this aspect, Gimenez et al. suggested that buildings can be represented as block models using footprint extrusion [55]. For the surrounding buildings, the features that influence the morphing procedure with the UWG are modeled by simply extruding their footprints. All buildings within the selected urban blocks are modeled as extruded blocks from their footprints as one of the urban geometry workflows supported by the Rhinoceros/Grasshopper/Dragonfly plugin, which is one of the Ladybug Tools software for translating geometry and CAD.

**Table 1.** Residential building regulation in Irbid. Source: Regulation No. (1) of 2022—Building Regulations and Organizing Cities and Villages (Prime Ministry of Jordan, 2022) [53].

| Residential Type | Setbacks (m) | | | BUP (%) | Floor No. | Height (m) | Main Facade Length (m) |
|---|---|---|---|---|---|---|---|
| | Front | Side | Back | | | | |
| Type A | 5 | 4 | 5 | 39 | 4 | 17 | 25 |
| Type B | 4 | 3 | 4 | 45 | 4 | 17 | 18 |
| Type C | 3 | 2.5 | 3 | 51 | 4 | 17 | 15 |

- Urban block scenarios: Due to the neglected parameters by the local regulations, the urban block designs have a random orientation and fixed heights (4 floors with 12 m) as well as a bigger range of data pertaining to the included parameters. The comparative study is divided into three steps: a comparative study between the different residential types (Type A, Type B and Type C), which is mainly represented by BUP; a comparative study between different scenarios for the same residential types; and a comparative study between the whole urban block designs parameters based on EUI and OTC. Therefore, the scenarios have been performed using three methods, as follows. (1) The orientation method: There are six main urban block designs that have been modeled according to their parameters in reality, while the scenarios are those six main urban block designs to be oriented east–west (90°) (long side), the second phase to be oriented north–south (0°), and the third phase to be oriented SE–NW (45°). Here, according to the local methods, the buildings are always perpendicular to the urban block layout (long side); therefore, each urban block design has four scenarios—main scenario, east–west or horizontal scenario, north–south or vertical scenario, and SE–NW scenario—to compare among all different types with the same orientation and to study the orientation effects on EUI and OTC. (2) The height method: This stage uses five different urban block designs, including different building types, to include a high range of CAR. (3) Distances between buildings: We obtained the distances between buildings from the setbacks, while the setbacks were not necessarily different among different types due to the main facade length restrictions; therefore, in this case, we set the scenarios to be restricted to the main facade length and scenarios to be restricted to the building setbacks.

*3.5. Weather Data Morphing*

3.5.1. Urban Weather Generator (UWG)

The UHI impact is affected not only by the building forms and urban fabric patterns, but also by anthropogenic activities, such as traffic, street lighting and building operation, which have an impact on the quantity of heat released to urban canyons.

A suitable approach to consider for such phenomenon is creating appropriate weather datasets through the adjustment (i.e., morphing) of existing weather data collected at rural sites and available in ".epw" format, which is convenient for several building simulation tools. UWG builds upon these premises and can estimate the hourly urban canopy air temperature and relative humidity considering the heat exchange and the air stratification at different atmospheric layers [18,35,56,57].

The meteorological variables obtained from the oppositenitial rural TMY file, urban morphology data should be provided to UWG to determine the buildings' average height, footprint density ratio (i.e., the ratio of the footprints of the buildings to the urban site area) and facade-to-site ratio (i.e., the ratio of the facades of the buildings to the urban area). These parameters are obtained by converting buildings' footprints derived from the Rhinoceros 3D model to the 3D model by using the DF Building from Footprint (BuildingFootprint) component, as shown in Figure 4.

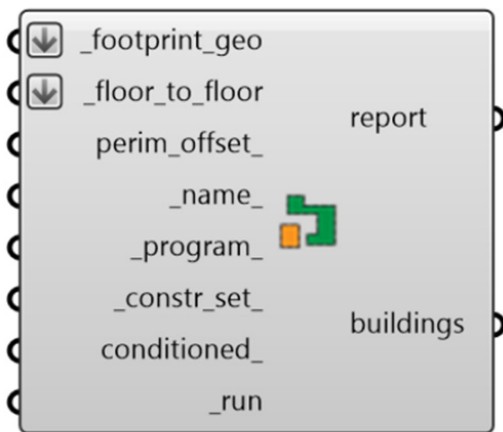

**Figure 4.** DF Building from the Footprint (BuildingFootprint) component. This is a figure. Schemes follow the same formatting.

For sensible heat fluxes emitted from heat sources within the urban area, the UWG separates the component due to vehicles, street lighting and pedestrian activity from that due to buildings. An example of these heat resources is the heat emitted by HVAC systems; therefore, the user must define both contributions separately. In addition, the absorbed shortwave solar radiation and the released longwave radiation calculations within the Urban Boundary Layer (UBL) must also be considered. These calculations can be performed by setting the thermal and optical parameters for the whole constructed surface with accurate values by a graphical interface under the Dragonfly tool within the Grasshopper environment, while the surrounding buildings have been defined as shading contexts, and the solar distribution module with full interior and exterior reflections is used.

These values have been obtained from the Jordanian heat transfer coefficient building codes [54,58,59], Jordan National Building Council [60], local material characteristics references [61], and site observation and validation with related studies [32,62–68]. Table 2 presents the different simulation parameters used in this study. We set the simulation parameters for the UWG component shown in Figure 5, which generates the "urban" epw file. UWG validation was based on the case of Boston, Basel and Toulouse, which was validated by Salvati et al. in the climate context of Barcelona and Rome with the same climate classification as Irbid [69]. Furthermore, the results have been compared and validated with the study of Ayyad in Amman, Jordan, approximately 78 km from the site location [70]. Figure 6 shows the hourly air temperature and relative humidity charts.

**Table 2.** EnergyPlus simulation parameters (according to Jordanian Building Codes [54,58–61,68]).

| Parameter | | Value [Residential] |
|---|---|---|
| **HVAC** | Heating/cooling setpoints | 21°/24° |
| | Schedule | Weekdays 16:00–24:00; weekends 07:00–24:00; sleeping 24:00–08:00 (heating from December–March, cooling from April–November) |
| | HVAC system | PTAC with baseboard electric |
| **Zone loads** | Lightning | 4 W/m² |
| | Schedule | Weekdays: 00:00 (25%)–06:00 (40%)–08:00 (15%)–16:00 (40%)–18:00 (150%)–23:00 (40%) weekends: 00:00 (50%)–11:00 (75%)–21:00 (100%)–23:00 (75%) |
| | Occupancy | 0.04 People/m² |
| | Schedule | Weekdays: 00:00 (50%)–06:00 (75%)–08:00 (25%)–16:00 (75%)–18:00 (100%)–23:00 (75%); weekends: 00:00 (50%)–11:00 (75%)–21:00 (100%)–23:00 (75%) |
| | Equipment | 5 W/m² |
| **Materials properties** | Walls | U = 1.32 W/m²K |
| | Roofs | U = 0.88 W/m²K |
| | Floors | U = 3.44 W/m²K |
| | Windows | U = 5.40 W/m²K |
| Window-to-floor ratio | | 15% |
| Infiltration | | 0.0003 m³/s-m² |

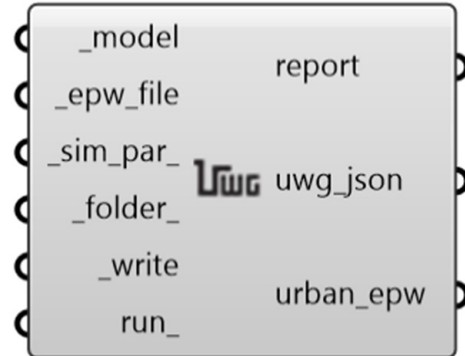

**Figure 5.** Urban weather generator component.

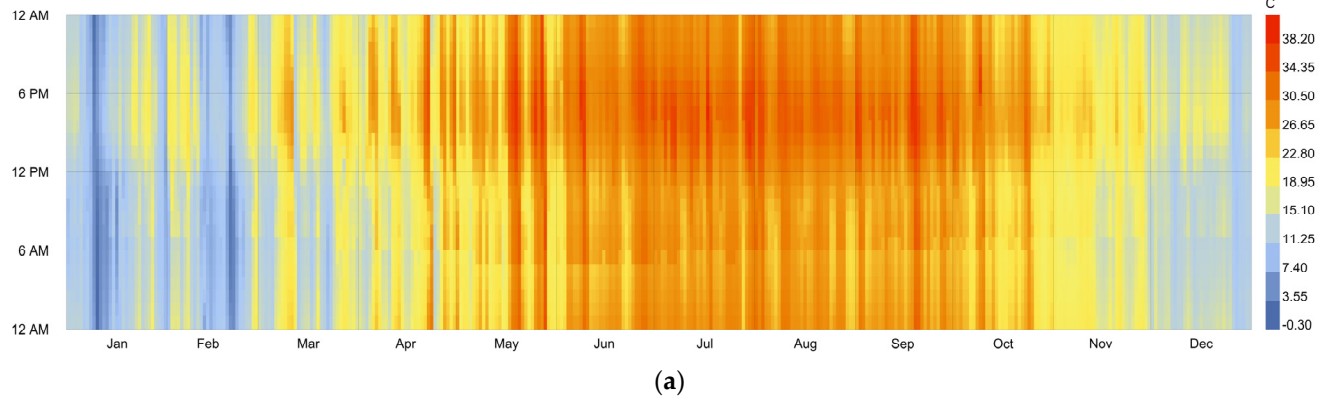

(a)

**Figure 6.** *Cont.*

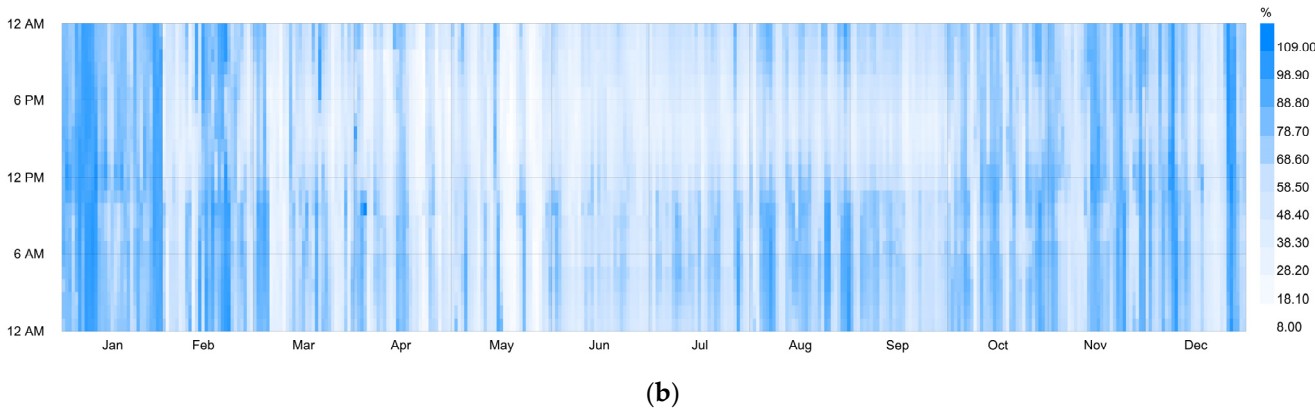

**(b)**

**Figure 6.** (**a**) Hourly air temperature chart; (**b**) hourly relative humidity chart.

### 3.5.2. Annual Wind Velocity Calculation

Wind is one of the most influential factors affecting outdoor thermal comfort in urban areas and outdoor spaces. Wind velocities and directions differ from place to place within the city. Small details can change the wind factors, therefore affecting outdoor thermal comfort calculations. A validated plugin of Eddy3d (Ver. 0.3.8.0) in Grasshopper/Rhinoceros that uses BlueCFD/OpenFOAM to calculate the annual wind speed.

Eddy3d uses the OpenFOAM blockMesh utility for the background mesh and snappy-HexMesh to snap the background mesh to the building geometry. A cylindrical simulation domain approach was used for the background mesh, which allows the same computational mesh to be reused for each wind direction, therefore reducing computation time and storage space [17,48]. Within the cylindrical mesh, the mesh within a refinement box that surrounds the buildings of interest was refined. Figure 7b shows that the inner refinement box was performed with OpenFOAM. Based on the best practices, we set up the simulation domain, considering all relevant surrounding buildings. For this study, 8 wind direction methods have been used in 45° intervals (0°, 45°, 90°, 135°, 180°, 225°, 270°, and 315°), as shown in Figure 7a.

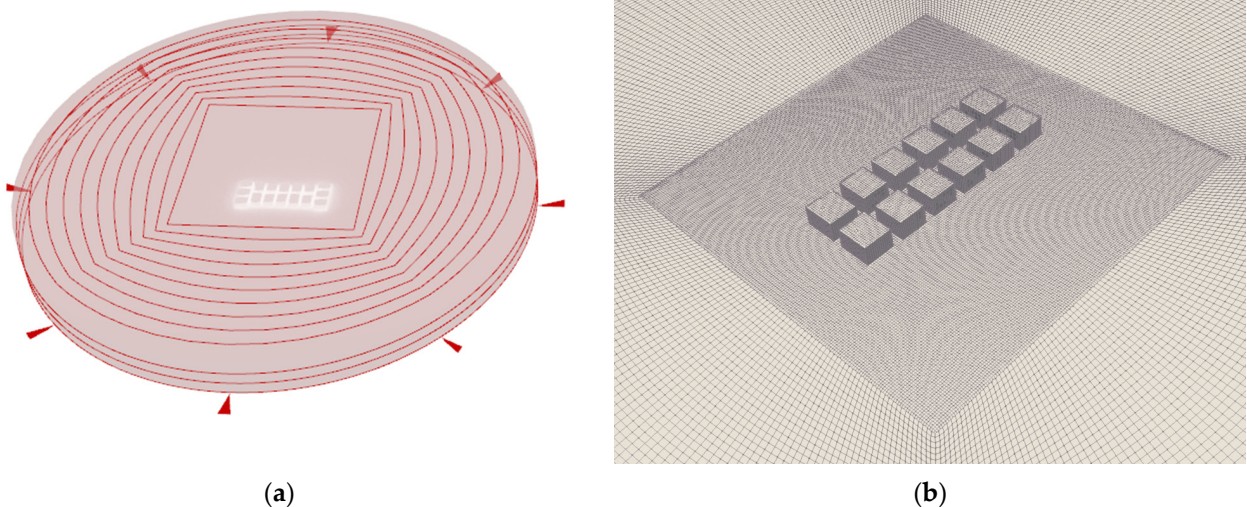

**(a)**        **(b)**

**Figure 7.** CFD simulation results: (**a**) resulting cylindrical simulation domain obtained with the Grasshopper/Eddy3d plugin (Ver. 0.3.8.0) with 8 wind directions; (**b**) resulting mesh obtained with OpenFOAM for the inner refinement box.

The simulated wind speed was derived from the annual weather data that is closest to the site location to be served by RANS simulations, where the simulated wind speeds represent the average velocity from that direction. All of the computational fluid dynamics

(CFD) simulations are run without trees to make all the urban blocks with the same scenario an equitable comparative study focusing on the urban block parameters as a result of the local buildings' regulations. Table 3 summarizes the CFD simulation settings used in this study.

**Table 3.** CFD simulations settings.

| Category | Variable | Value |
| --- | --- | --- |
| Boundary conditions | Wind directions | (0°, 45°, 90°, 135°, 180°, 225°, 270°, and 315°) |
| | $U_{ref}$ | Average |
| | $Z_{ref}$ | 10 m |
| | $Z_0$ | 1 m for a typical urban area |
| | $Z_{Ground}$ | 0 m |
| Simulation domain (cylindrical) | Block size<br>Inner rectangular size<br>Outer radius size<br>Height | Best practice value |
| Mesh settings | Accuracy of building mesh | 3 |
| | Accuracy of building features mesh | 3 |
| | Accuracy of bounding box mesh | 0 |
| | Accuracy of ground mesh | 3 |
| | Mode | With snapping, no layers |
| Run settings | Number of iterations | 3000 |
| | Turbulence model | k-Omega SST |
| | Relaxation factors | Optimized |
| | Solution and algorithm control optimized | Optimized |

In the CFD simulations for all cases, wind velocity plots were extracted at the height of average human height in Jordan (Av. height = 1.66 m), the same location as the MRT calculations. For CFD simulation validation, we applied our cylindrical simulation domain and other CFD simulation settings to a simple case provided by the Architectural Institute of Japan benchmark that has been performed by the wind tunnel method [71]. Figure 8 shows the results of CFD simulation method validation against the wind tunnel method ($R^2 = 0.95$).

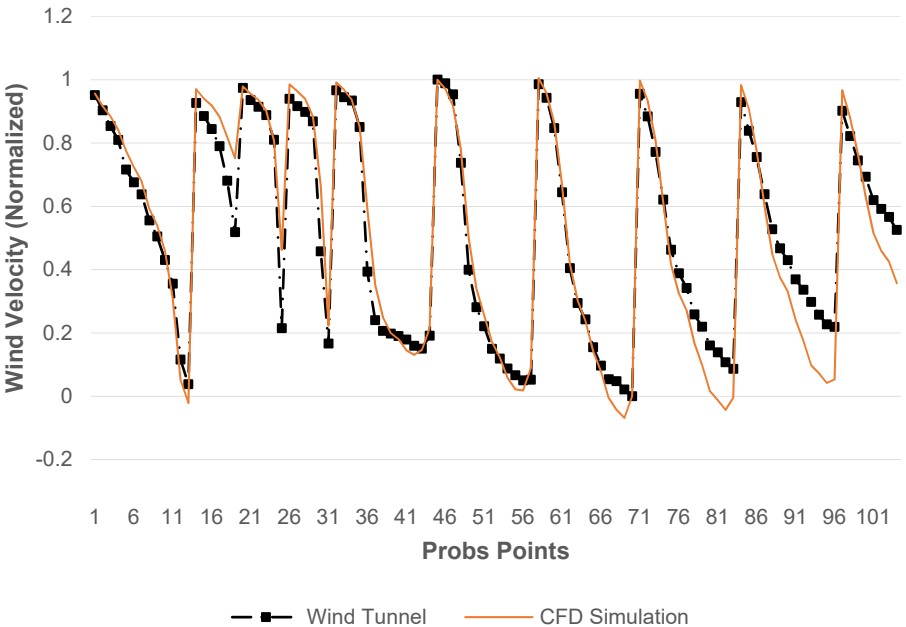

**Figure 8.** CFD simulation method validation against the wind tunnel method.

### *3.6. Performance Evaluation*

### 3.6.1. Energy Use Intensity (EUI)

To evaluate the EUI of the urban buildings, we used the DF Model to Honeybee component (Ver. 1.4.0) under the Dragonfly engine as one of Ladybug's tools. This component converts a Dragonfly Model used in UWG with the same simulation parameters (see Section 3.5.1) into a series of Honeybee Models to be exported to OpenStudio by using the HB Model to OSM component, then run through EnergyPlus to obtain the results of energy use simulations as EUI. Another way is by using the DF Model to geoJSON component, which converts a Dragonfly Model into an URBANopt-compatible geoJSON with linked Honeybee Model JSONs to run the URBANopt component to obtain the energy use simulation results as EUI, and the two ways gave us the same results. To make comparisons with other scenarios more appropriate, the heating and cooling energy consumptions for each urban block scenario were normalized in relation to the total area. As a result, the performance evaluation metrics used to analyze energy use were calculated as annual kWh. In addition, the calculations of monthly and total energy use for both cooling and heating have been done in this study.

### 3.6.2. Outdoor Thermal Comfort (OTC)

In the last few decades, there have been numerous proposed outdoor thermal comfort metrics, many of which have been built to consider such climatic factors as relative humidity, wind speed, or direct sun. Hence, it is critical to select the most appropriate and comprehensive outdoor thermal comfort metric for this study. The universal thermal climate index (UTCI) was chosen for this study. In the last few years, in order to assess the comfort of outdoor thermal environments, one of the most commonly used thermal indices has been UTCI [72]. UTCI proved to be much more useful for describing the physiological comfort of the human body under specific meteorological conditions [73]. In addition, related to the human body, the UTCI is very sensitive to adjustments in ambient stimuli. UTCI describes the temporal variability of thermal conditions better than other indices. Moreover, UTCI is capable of expressing even slight variations in the intensity of meteorological stimuli [73]. UTCI requires four variables to be calculated: air temperature, wind speed, relative humidity, and MRT. Besides that, UTCI considers the clothing adaptation of the population in response to actual environmental temperatures. All other different factors, together with age, height and weight, are averaged over the population. UTCI has 10 thermal stress categories that match the specific human physiological responses to the thermal environment, as explained in Table 4.

**Table 4.** UTCI equivalent temperature categorized in terms of thermal stress.

| UTCI (°C) | Thermal Stress Classification |
|---|---|
| Above +46 | Extreme heat stress |
| +38 to +46 | Very strong heat stress |
| +32 to +38 | Strong heat stress |
| +26 to +32 | Moderate heat stress |
| +9 to +26 | No thermal stress |
| 0 to +9 | Slight cold stress |
| −13 to 0 | Moderate cold stress |
| −27 to −13 | Strong cold stress |
| −40 to −27 | Very strong cold stress |
| Below −40 | Extreme cold stress |

With the UTCI Comfort Map component under Grasshopper/Honeybee-Energy (UTCIMap) (Ver. 1.4.0), we can calculate UTCI, heat, and cold stress conditions and visualize them as a thermal stress map.

With the microclimatic wind speed, MRT, air temperature and relative humidity calculations, the HB UTCI comfort map component uses EnergyPlus to obtain the surface

temperatures of the ground and surrounding facades. This component uses EnergyPlus to calculate longwave radiant temperatures through spherical view factors from each sensor with a 1.66 m height set (wind speed's probe point height) to the model's room surfaces. Then, these view factors are multiplied by the surface temperatures output by using EnergyPlus to obtain the longwave MRT at each sensor. For the outdoor sensors, the sky view of each sensor is multiplied by the EPW sky temperature to account for the longwave radiative exchange with the sky. While shortwave MRT is calculated with the radiance-based enhanced 2-phase method, which represents the direct sun by tracing a ray from each sensor to the solar position.

Solar body parameters include body posture (standing), SHARP represents the solar horizontal angel relative to front of person (135), the angel between the sun and person face (always facing from the sun), and the absorptivity refers to the average shortwave absorptivity of the body including clothing and skin color (0.7 refers to brown skin). After we set the whole input parameters, we can run HB UTCI thermal map to obtain four important thermal maps: Average UTCI map, Thermal Comfort Percentage (TCP) map, Heat Sensation Percent (HSP) map, and Cold Sensation Percent (CSP) map.

## 4. Results

### 4.1. Outdoor Thermal Comfort Distribution: Vertical and Horizontal Scenarios

According to the summer results, the N–S street canyons have the lowest Av. UTCI while the outer setbacks on the south side have the highest Av. UTCI. E–W canyons have an Av. UTCI equal to or lower than the south outer setbacks, but higher than the rest. As Av. TCP, E–W canyons have the lowest Av. TCP and west setbacks have the highest Av. TCP, while N-S canyons have an Av. TCP higher than E–W canyons and lower than the outer setbacks. During the day, outer setbacks and E–W canyons are hotter than N-S canyons. East setbacks are exposed to short solar radiation during the morning and shaded after mid-afternoon, whereas west setbacks are shaded from morning until afternoon. South setbacks and E–W street canyons are fully exposed to solar radiation during the whole day, but E–W canyons depend on the height-to-width ratio. During the night, wider street canyons represented by outer setbacks are colder than the narrow street canyons represented by inner setbacks or distances between buildings because of the longwave radiation effect during the night, while outer setback areas can get rid of longwave radiation. In addition, the wind speed at the wide street canyons (outer setbacks) is faster than the wind speed within deeper street canyons between buildings; therefore, outer setbacks have an Av. HSP lower than the Av. HSP within the inner setbacks. The difference in Av. TCP between the outer and inner setback zones during the night is higher than the difference during the day. It is important to find a balance in urban block designs that meets the OTC between the day and night, e.g., the activity distribution within N–S canyons during the day and within the outer setback zones during the night.

In the winter season, the results are dominated by the highest Av. TCP within N–S street canyons, followed by E–W street canyons, where the lowest Av. TCP was within the south setback areas. Inner setback areas are more comfortable than the outer setback areas because they have a lower wind speed and better exposure to solar radiation, which is high during high solar angle altitudes and low during low solar angle altitudes, while outer setback areas are more exposed to solar radiation with a higher wind speed. In terms of annual Av. TCP, the results indicate that N–S canyons have the highest Av. TCP, while the south setback zones have the lowest Av. TCP among all zones. The north and west setback areas are more comfortable than the E–W canyons and east setbacks, as shown in Figure 9. There is a high correlation between Av. TCP within N–S canyons and the entire urban blocks Av. TCP, as shown in Figure 10, which indicates that N–S canyons have the highest impact on OTC within the urban block designs. Increasing the N–S canyons while decreasing or controlling the south setback zone, east setback zone and E–W canyons, and increasing the north and west setback zones will increase the Av. TCP of the urban block. In this regard, increasing the facade facing west and east or the N–S building axis means

increasing the N–S street canyon length and, therefore, enhancing the OTC, which is quite the opposite with regard to the facade facing north and south, or the E–W building axis. However, these zones must be studied with CAR, as will be discussed (see Section 4.3). Regarding Av. TCP within west setback zones, which is higher than within east setback zones, east setbacks and building facades are fully exposed to solar radiation during the morning and shaded from afternoon to evening, while quite the opposite is true for the west setback zone. The difference between the east and west sides during the night is that since the prevailing wind direction in Irbid is west (270°), the wind formulates a turbulent wake zone at the west side after crossing the adjacent urban block or as a wind flow separation point, which decreases the air temperature, while the east side will be within the cavity zone, which means that the wind speed within the east side is higher than the wind speed within the west side. Wind speed can decrease the air temperature and accelerate the longwave radiation transferring to the sky vault, which reduces the heat sensation.

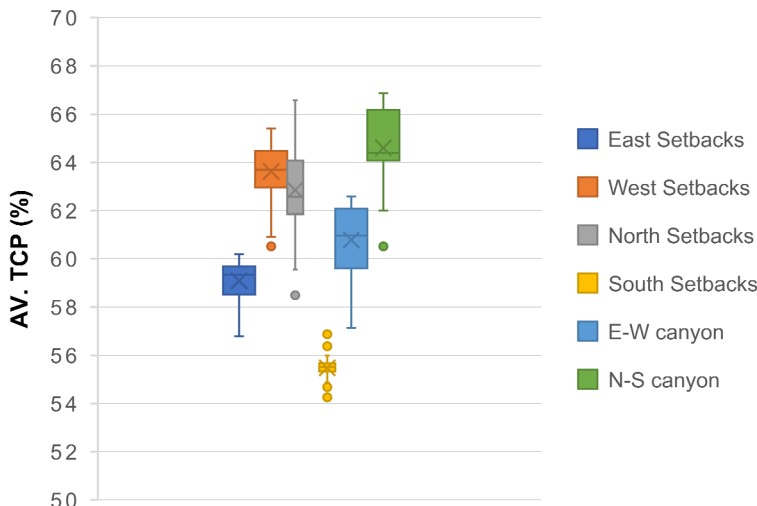

**Figure 9.** Box and whisker plot of annual Av. TCP within different urban block zones.

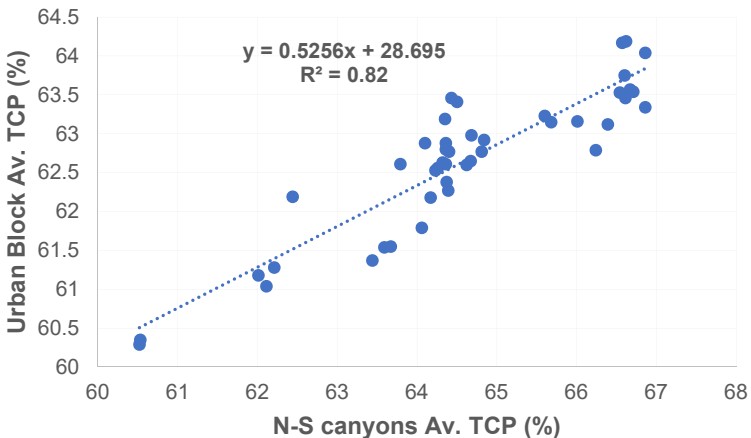

**Figure 10.** The correlation between annual Av. TCP within the N–S canyon and urban block total Av. TCP.

The difference between winter and summer is that in the winter, there is a need to increase the Av. UTCI to decrease the Av. CSP by exposing to solar radiation and decreasing the wind speed, while in the summer, it is quite the opposite. In the summer, the Av. UTCI can be decreased by creating narrow street canyons to be shaded during the day, but during the night, the narrow street canyons have a lower wind speed than the wider street canyons. To achieve a balance between different needs, urban blocks should be divided into narrow street canyons and wide street canyons (compacted urban blocks surrounded by wide street canyons). Compacted urban blocks have the advantage of enhancing a microclimate but

have a negative impact by slowing down and blocking wind [36]. Blocking and slowing down wind during the winter reduces the Av. CSP, as shown in Figure 11a and, hence, increases the Av. TCP. Narrow street canyons are more comfortable during the daytime in the summer, while in the winter, they are more comfortable during the day by being exposed to solar radiation and with low wind speeds during the night. Wide street canyons are more comfortable during the night in the summer and during the day in the winter. The results highlighted that the decreasing the summer Av. UTCI within the whole outer and inner zones increases its annual Av. TCP. In total, decreasing the Av. UTCI in the summer increases the annual Av. TCP, as shown in Figure 11b.

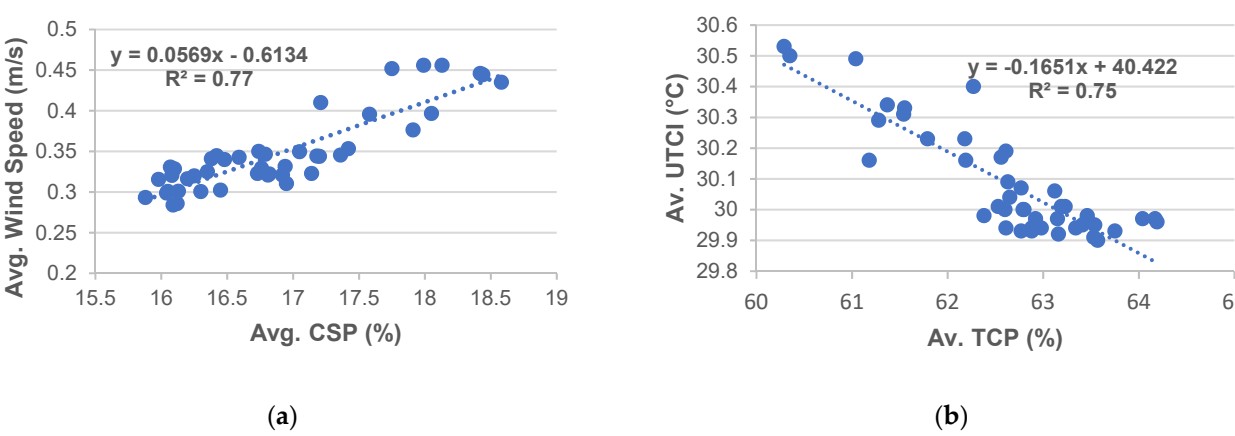

(**a**)                                                                                  (**b**)

**Figure 11.** (**a**) The correlation between winter Av. CSP and Av. wind speed; (**b**) the correlation between summer Avg. UTCI and annual Av. TCP.

*4.2. The Effect of Urban Block Design Orientation*

4.2.1. The Correlation between Outdoor Thermal Comfort (OTC) Urban Block Design Orientation

Since we discussed the OTC distribution in vertical (buildings' orientation E–W) and horizontal (buildings' orientation N–S) scenarios in (see Section 4.1), oriented urban blocks towards NW–SE or NE–SW influence the OTC distribution. NW–SE outer setbacks have an Av. TCP lower than those of the north setbacks in the vertical scenario and higher than the east setbacks in the horizontal scenario, whereas NW–SE on the opposite side has an Av. TCP lower than the west side Av. TCP in the horizontal scenario and higher than the south side Av. TCP in the vertical scenario. NE-SW sides, the upper one with an Av. TCP higher than the north side Av. TCP in the horizontal scenario and the west side Av. TCP in the vertical scenario, whereas the opposite NE–SW side is higher than the south side Av. TCP in the horizontal scenario and lower than the east side Av. TCP in the vertical scenario. The urban block has been oriented from 0 to 45 or 90 to 45, decreasing the Av. UTCI within E–W canyons, whereas it witnessed an increase in the Av. UTCI within N–S canyons. Orienting the urban block layouts and buildings towards NW–SE or NE–SW is better for OTC than vertical urban blocks with buildings oriented towards E–W as long as the positive gap in Av. TCP within inner setback zones is higher than the negative gap within outer setback zones, and vice versa for the comparative between ordinal scenarios and horizontal urban block scenarios. It is a matter of proportion. According to the results, the highest Av. TCP was achieved in urban blocks with a NE–SW orientation, while the lowest was achieved with the same orientation as shown in Figure 12. The urban blocks oriented towards E–W (87–90°) with buildings oriented towards N–S (357–0°) recorded the highest Av. TCP median. The urban block design method in Irbid creates urban block orientations perpendicular to building orientations. It is not necessarily true that urban blocks with ordinal orientations are more comfortable than urban blocks with cardinal orientations. It is crucial to understand the spatial distribution of OTC and its correlations among both wide and narrow streets, as in our study.

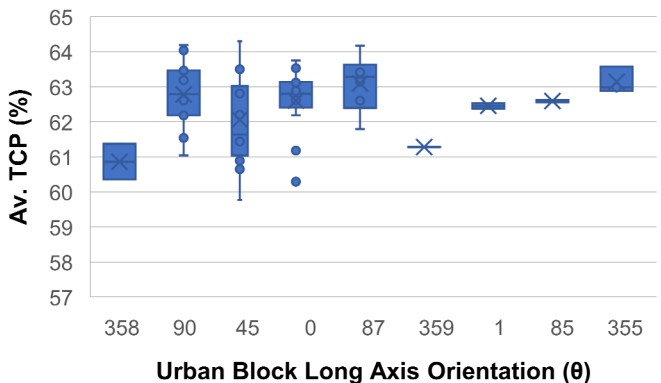

**Figure 12.** Box and whisker plot of annual Av. TCP by different orientations.

4.2.2. The Correlation between Energy Use Intensity (EUI) and Urban Block Design Orientation

Walls facing south have the highest solar gain. In contrast, walls facing north tend to be unexposed to solar radiation, and east-facing and west-facing walls gain some solar radiation during the morning and afternoon, respectively. Walls facing east and west could have shade potential due to adjacent buildings' count on the aspect ratio, which enhances the building's performance in cooling energy. On the other side, walls facing south are more affected by the shading amount during the winter, resulting in increased building heating demand. Consequently, urban blocks with N–S long-axis buildings have superior performance in terms of cooling energy demand. This reminds us that increasing the N–S building axis at the expense of decreasing width will enhance both OTC (see Section 4.1) and EUI. The results shown in Figure 13 illustrate that E–W (87–90°) urban blocks contain buildings in the N–S orientation with more frequency in achieving the lowest EUI than the other orientation, whereas there is more frequency with the highest EUI in (45°), where the lowest EUI has been achieved with an orientation of (90°).

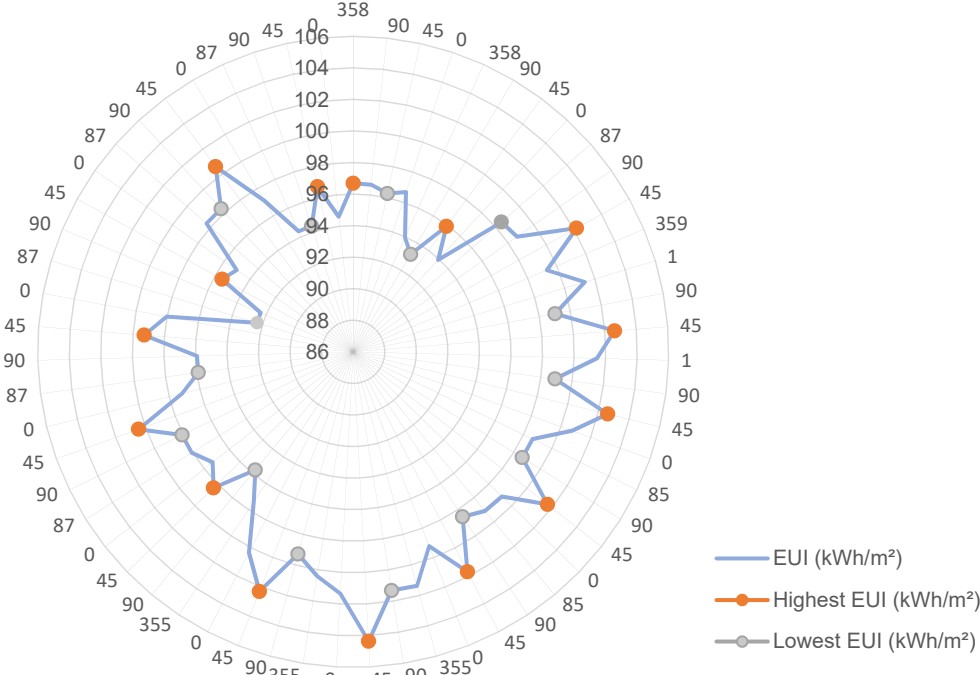

**Figure 13.** Urban block scenario energy use performance evaluation with orientation.

### 4.2.3. The Combined Performance of Outdoor Thermal Comfort and Energy Use Intensity

In order to ascertain the preferable environmental performance orientation, urban block orientations are compared based on the two combined performance indicators: EUI and Av. TCP. The objective function [32,74] attempted to distinguish the outcome that has the best fitness of both Av. TCP and EUI for all urban block scenario results. The goal is to increase the Av. TCP and decrease EUI; therefore, the EUI part of the equation is multiplied by −1. The higher the fitness function (y), the greater the environmental performance. Equation (1) gives the calculation of the fitness function:

$$y = (\text{Av. TCP}_i - \text{Av. TCP}_{min}).C_1 + -1(\text{EUI}_i - \text{EUI}_{min}).C_2 \tag{1}$$

$$C_1 = \frac{100}{(\text{Av. TCP}_{max} - \text{Av. TCP}_{min})}, \; C_2 = \frac{100}{(\text{EUI}_{max} - \text{EUI}_{min})}$$

where i is the iteration result, min is the minimum value of all iterations and max is the maximum value of all iterations.

The outcomes highlighted that E–W urban block design 13 has the highest fitness function value. All the E–W urban block scenarios dominated with the highest fitness function in comparison with the other scenarios, except twice. The first one combined the orientation of two buildings, which makes sense. The other case had an 87° orientation related to the aspect ratio, which will be discussed (see Section 4.3), whereas urban blocks with 45° orientations have the worst fitness function value. Urban blocks with 87° showed satisfactory environmental performance, but since a total of E–W urban blocks with N–S buildings (0°) are the most favorable, our study will, therefore, adopt the cardinal orientations as the fundamentals of the urban block design proposal, since the building's orientation is perpendicular to the orientation of the urban block layout as a design method in the study area.

### 4.3. The Effect of Canyon Aspect Ratio (CAR)

### 4.3.1. The Effect of Canyon Aspect Ratio (CAR) on Energy Use and Outdoor Thermal Comfort (OTC)

Both Av. UTCI and EUI have a negative correlation with CAR. Regarding the objective function, the results in Figure 14 manifest that the increase in CAR increases the objective function, which means enhancing the environmental performance. Nevertheless, EUI has no indicators of the range of CAR, contrary to Av. UTCI. Additionally, as the distance continues to decrease, EUI continuously decreases until connected to the adjacent buildings' walls, which means increasing the shade on the building walls and, therefore, decreasing cooling energy and increasing heating energy. This resulted in a preference for increasing the CAR over decreasing it, due to the higher effect of cooling energy over heating energy. Decreasing E–W CAR will increase the amount of shadow on a wall facing south, resulting in increased heating energy.

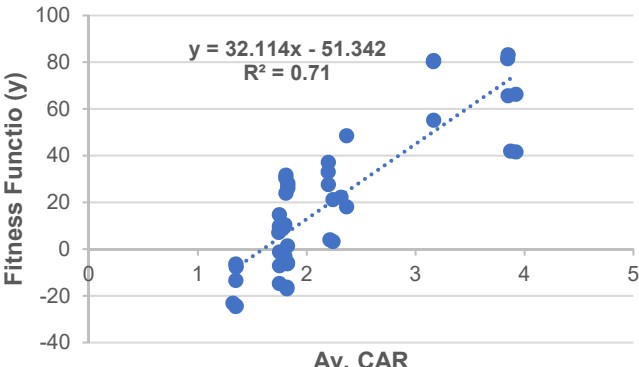

**Figure 14.** Correlation between the fitness function and Av. CAR.

Increasing E–W CAR with a wall facing south will increase cooling and heating energy due to the difference in the solar angle altitude between summer and winter. To find the CAR range, we will rely on the Av. UTCI results to be more objective, aiming to create a balance in the environmental performance between energy use and OTC. The results also demonstrate that increasing the height of the buildings enhances the energy performance of the buildings when comparing urban block designs with different heights derived from the same urban block design with fixed setbacks.

### 4.3.2. The Correlations between Outdoor Thermal Comfort (OTC) and Canyon Aspect Ratio (CAR)

The west and north CAR, in addition to the N–S and E–W CAR, have a negative correlation with Av. UTCI. The east and south sides, as we mentioned (see Section 4.1), have the worst Av. UTCI and it is slightly different from the other CAR. The east side is considered the second side of the E–W wide street canyon, with the north side being fully exposed to solar radiation during the day and suffering from longwave radiation during the night. There is a negative correlation between H/east setbacks and Av. UTCI. Increasing the south setbacks will increase the distance between the north and south sides, which will create more uncomfortable areas. In addition, increasing the building's © and height will negatively affect the OTC. The south setbacks are affected by longwave radiation, which is highly concentrated next to building walls. We suggest increasing the setbacks until a 2 m point, and the increase above should be planted. This study emphasizes that it is important to decrease the setbacks on the south side, but not less than 2 m, and decrease the building's height and ©, aiming to reduce the uncomfortable areas on the south side. Creating green belts or areas with terraced floors or green facades and floors can help. Regarding the east sides, these areas are exposed during the morning, but suffer from a lack of night breeze in compacted urban blocks. Creating green areas or belts with green facades or terraced floors with greening can improve the OTC. It is important to emphasize that terraced floors should be studied side by side with EUI.

The north and west sides or setbacks, N–S street canyons, and E–W street canyons have similar correlations with the CAR, as shown in Figure 15. All these zones clearly indicate that the Av. UTCI started decreasing after CAR = 1.5, and is not highly changeable with CAR > 4, except for the west setbacks (CAR > 4.5) and E–W street canyons with CAR > 3.5. Increasing CAR in N–S street canyons means decreasing the exposure to solar radiation duration as well as west setbacks and increasing the shaded areas, therefore, increasing CAR more than 4 in these areas will not be noticeable in the decrease in Av. UTCI since N–S street canyons will be fully exposed at 11:00 am or 12:00 pm, while west setbacks will be fully exposed after that time. E–W street canyons and north setbacks, the decreasing of CAR in these areas means decreasing the exposure to solar radiation by increasing the amount of shadow. E–W street canyons will be fully shaded with CAR = 3.5, and increasing the CAR more than 3.5 will be negatively affected by longwave radiation, whereas north setbacks with CAR = 4 have the same Av. UTCI of (CAR > 4).

However, we should keep in mind the difference in Av. UTCI between the whole zones with the same CAR as we mentioned (see Section 4.1). We recommend that the need for planting increases with the Av. UTCI difference among the zones (south setbacks > E–W street canyons > east setbacks > north setbacks > west setbacks > N–S street canyons) and the opposite is true when the setbacks increase. The increase in CAR should be accompanied by an increase in the building height, with an exception for the south setbacks.

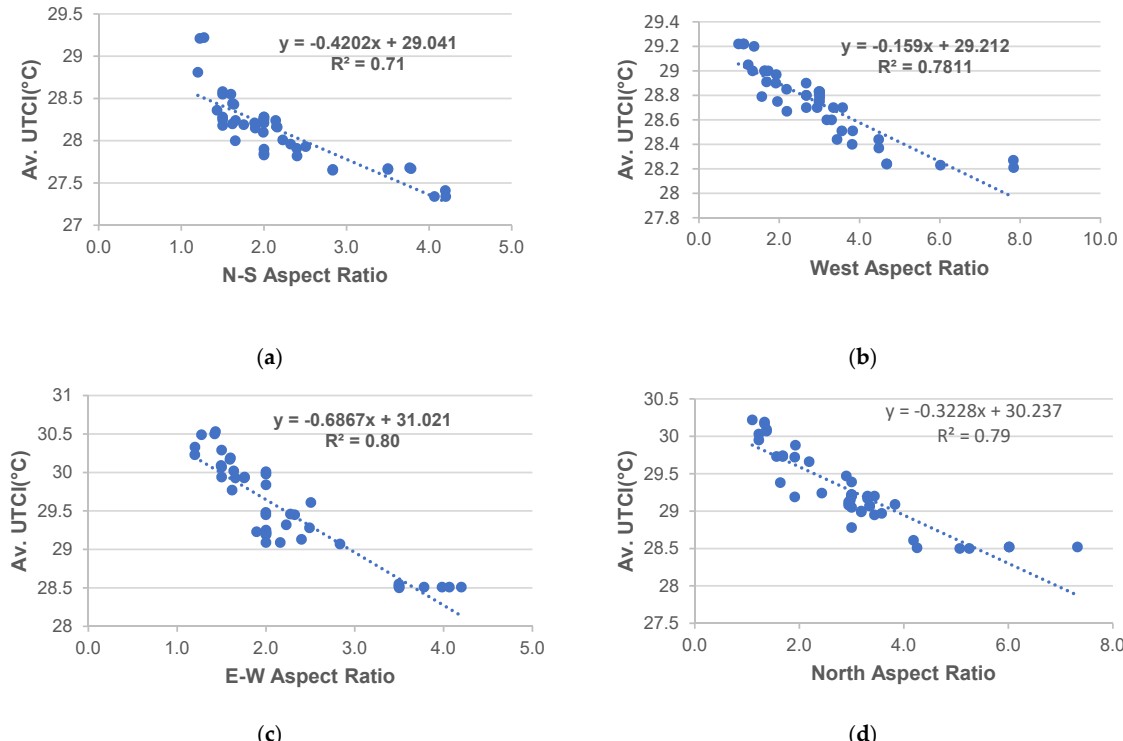

**Figure 15.** (**a**) Correlation between summer maximum Av. UTCI and N–S aspect ratio; (**b**) correlation between summer Av. UTCI and west aspect ratio; (**c**) correlation between summer Av. UTCI and E–W aspect ratio; (**d**) correlation between summer Av. UTCI and north aspect ratio.

### 4.4. Built-Up Percentage (BUP) Effect

According to Regulation No. (1) of 2022—Building Regulations and Organizing Cities and Villages (Prime Ministry of Jordan, 2022) [53], the definition of the built-up percentage is the ratio of the area of the largest horizontal floor to the plot area. Decreasing the BUP decreases the distance between buildings as well as the CAR, which will negatively affect the fitness function according to the results (see Section 4.3.1), resulting in decreased environmental performance. The results shown in Figure 16a demonstrate that urban blocks with higher BUP have a higher fitness function value. In terms of OTC, the results shown in Figure 16b prove the same correlation with Av. TCP. Regarding BUP and energy use, decreasing the BUP will enhance the cooling performance during the day, while enhancing the heating performance during the night; however, increasing the BUP decreases the daylight potential. Thus, increasing BUP is more favorable than decreasing it.

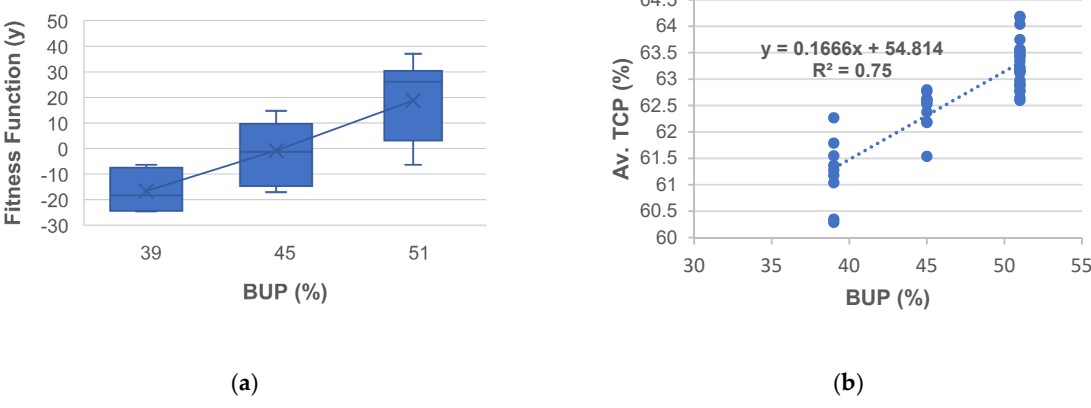

**Figure 16.** (**a**) Box and whisker plot of the fitness function value within different built-up percentages (BUP); (**b**) correlation between annual Av. TCP and built-up percentage (BUP).

### 4.5. Urban Block Design Proposals

Based on the achieved results, we applied new parameters to the urban block with the worst environmental performance in the urban block design number 3 horizontal scenario to be compared with the urban block number 13 horizontal scenario with a higher environmental performance. Figure 17 shows UTCI maps for these urban block designs, whereas Table 5 illustrates that the parameters used for achieving the results have higher environmental performance compared to all urban blocks tested in this study. The difference between optimized urban block designs 1 and 2 is in N–S, with CAR = 4.2 and CAR = 4, respectively. The results demonstrate that increasing N–S (CAR) by more than 4 will decrease EUI, especially cooling energy, but will negatively affect the Av. TCP because of longwave radiation during the night, and, therefore, increasing HSP will no longer decrease the Av. UTCI after CAR = 4. It was observed that east and south setbacks should be decreased, while on the other hand, north and west setbacks can be increased. Table 6 points out a set of recommendations for Building Regulations and Organizing Cities and Villages. The maximum height-to-width ratio among all zones means that at these points, it is recommended to plant with the height-to-width ratio decreasing. South setbacks are the most uncomfortable areas and must be designed with more greening than the others. Setbacks can be increased to the point of the minimum CAR but must be compensated by greening. N–S canyons could be enhanced without greening or could be enhanced with greening but lower than the other zones. It is necessary to tend more to the most uncomfortable zones.

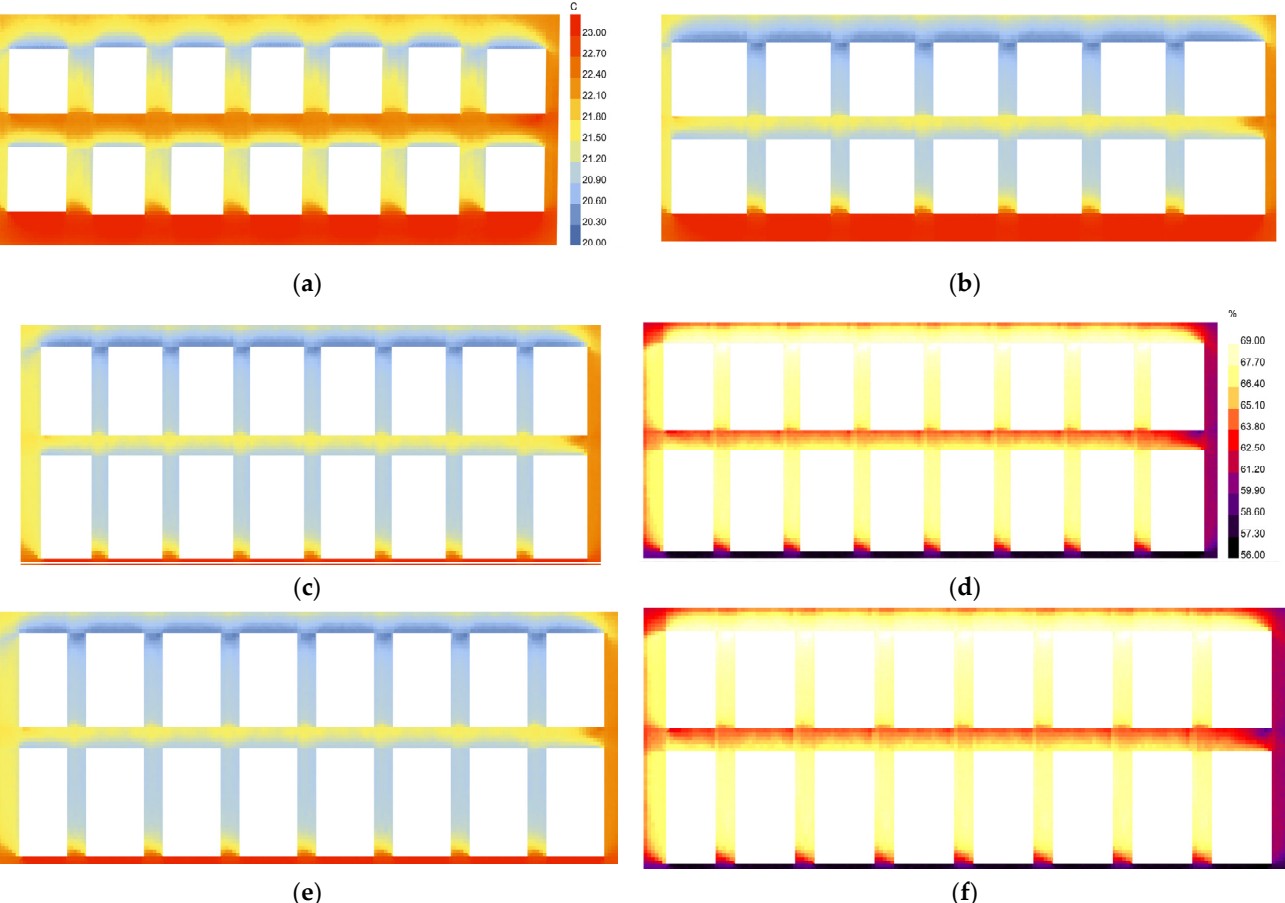

**Figure 17.** (**a**) Urban block design No. 3 horizontal scenario annual UTCI map; (**b**) urban block design No. 13 horizontal scenario annual UTCI map; (**c**) optimized urban block design No. 1 annual UTCI map; (**d**) optimized urban block design No. 1 annual TCP map. (**e**) optimized urban block design No. 2 annual UTCI map; (**f**) optimized urban block design No. 2 annual TCP map.

**Table 5.** Comparative analysis between urban block designs within the study area and optimized urban block designs based on results.

|  |  | Urban Block Design No. 3 | Urban Block Design No. 13 | Opt. Urban Block Design No. 1 | Opt. Urban Block Design No. 2 |
|---|---|---|---|---|---|
| Parameters | Av. N–S (CAR) | 1.5 | 4.2 | 4.2 | 4 |
|  | Av. E–W (CAR) | 1.2 | 3.5 | 3.5 | 3.5 |
|  | East H/setbacks | 3 | 7.3 | 5.25 | 5.25 |
|  | West H/setbacks | 3 | 7.8 | 3.5 | 3.5 |
|  | North H/setbacks | 1.2 | 2.9 | 3.5 | 3.5 |
|  | South setbacks (m) | 9.8 | 7.17 | 2 | 2 |
|  | Building orientation | 1° | 0° | 0° | 0° |
|  | Block layout orientation | 90° | 90° | 90° | 90° |
|  | Av. building length (m) | 20.2 | 19.83 | 28 | 28 |
|  | Av. building width (m) | 16.57 | 18 | 15.88 | 15.68 |
| Results | EUI (kWh/m$^2$) | 98.694 | 92.36 | 90.997 | 91.38 |
|  | Cooling energy (kWh/m$^2$) | 33.253 | 28.838 | 28.109 | 28.32 |
|  | Heating energy (kWh/m$^2$) | 8.111 | 8.696 | 8.37 | 8.404 |
|  | Annual Av. TCP (%) | 61.55 | 63.54 | 64.93 | 65.03 |
|  | Summer Av. TCP (%) | 32.44 | 32.41 | 32.36 | 32.55 |
|  | Summer Av. HSP (%) | 67.56 | 67.59 | 67.64 | 67.45 |
|  | Winter Av. TCP (%) | 79.88 | 81.35 | 82.65 | 82.54 |
|  | Winter Av. CSP (%) | 17.99 | 16.58 | 16.06 | 16.21 |

**Table 6.** Urban block design recommendations for the optimum environmental design thresholds.

| Parameter | Value |
|---|---|
| Av. N–S (CAR) | $\geq$1.5–$\leq$4 |
| Av. E–W (CAR) | $\geq$1.5–$\leq$3.5 |
| East H/setbacks | $\geq$1.5–$\leq$4 |
| West H/setbacks | $\geq$1.5–$\leq$4.5 |
| North H/setbacks | $\geq$1.5–$\leq$4 |
| South setbacks (m) | $\geq$2–(minimized as possible) |
| Building orientation | 0–3° |
| Building length/width ratio | $\geq$1.2 |

## 5. Discussion

Aspect ratio is affected by building heights and built-up percentage (BUP), and this study demonstrates that aspect ratio has a negative relationship with both EUI and OTC, since it tested the effect of different building heights with fixed distances between buildings represented by setbacks. Regarding EUI, increasing building heights and BUP increases the percentage of shaded portions of the building facades, which exactly confirms the results of previous research conducted on residential building types in the city of Amman in Jordan [28], and decreases the cooling energy, which has a greater impact on building energy efficiency since Jordan is as close as it can be to the summer peak and results in increasing building heights and BUP decreasing EUI. These obtained confirmations confirm the findings of previous studies that EUI has a negative correlation with BUP, building heights, CAR and FAR [75,76]. However, somehow our findings differed from previous studies, possibly due to using different simulation settings that resulted in a positive correlation between FAR, building heights and EUI [32]. While our findings regarding OTC confirm the findings of this study that there is a negative correlation between building heights and OTC due to its effect on CAR [32], they also confirm the findings of previous studies [36,77]. In this context, compacted urban designs with higher CAR have a significant impact on environmental performance.

N–S street canyons have a lower Av. UTCI and it seems that street canyons in this orientation are more desirable for OTC, which matches the outcomes of the previous studies that recommended street canyons be oriented towards N–S, where E–W experience the worst thermal conditions [36,78,79]. Based on these findings, we concluded that increasing N–S proportions within the urban block design increases the Av. TCP and these proportions can be increased by building orientations and increasing the length of facades facing west and facades facing east, while E–W street canyon proportions can

be decreased by decreasing the length of facades facing north and facades facing south. That has been confirmed with previous studies where controlling and understanding the location distribution of open spaces formed by buildings' distributions can help enhance the OTC [16]. In terms of EUI, building shapes and orientations play a vital role in building energy demand, with the findings of this study demonstrating that buildings with N–S orientations have more benefits in energy performance, where increasing the length/width proportion in this orientation will decrease the energy demand because it reduces the cooling energy, since it decreases the solar gain amount at south-facing walls. On the other hand, buildings with ordinal orientations have the worst energy performance, and this converges with previous studies that concluded that N–S buildings have more energy efficiency than E–W, NE–SW and NW–SE within Mediterranean-climate regions [80]. This evidence is centered around N–S buildings having higher environmental performance regarding both EUI and OTC.

In compacted urban block designs, narrow and deep canyons are more comfortable than wide street canyons during the daytime; however, during the night in the summer, this study found that the Av. UTCI was lower within the inner setbacks (narrow and deep street canyons), while the Av. TCP was higher within the outer setbacks (wide street canyons). Therefore, surrounding compacted urban block design by wide street canyons represented by outer setbacks maintained as comfortable zones during nighttime will find the balance of space distribution between daytime and nighttime and the opposite exactly in the winter, as has been encouraged by a previous study that found that canyons with better thermal conditions during the day have higher temperatures during the night. Additionally, it is crucial to think of a compromise solution in terms of the design of urban canyons [81]. As a result, residential Type C has better environmental performance than Types B and C.

In terms of morphing weather variables, the morphing procedure does not modify weather variables such as the global horizontal solar irradiance and the wind velocity, which are then preserved from the initial rural TMY weather file. Although the first hypothesis is passable, since the solar irradiance available on a horizontal, unimpeded surface does not modify considerably from a rural to an urban site nearby, the second clearly brings an accurate result because the wind pattern certainly changes in urban environments. This weakness affects the accuracy of energy use estimation due to the inaccuracy of natural ventilation calculations, but it can be solved for OTC through wind simulations. This precisely concurs with a previous study related to future urban weather files, where the URBVENT project has proposed an algorithm to calculate hourly urban wind attenuation in urban canyons to be used for urban energy use estimation [82].

## 6. Conclusions

This study adopts a performance-driven approach that uses the capabilities of Ladybug Tools simulation workflow through Grasshopper/Rhin3D. This approach was applied to ascertain the correlations between different urban block design parameters (canyon aspect ratio, buildings and street orientation, facade length, street length, building' forms restricted to square and rectangle shapes, and built-up percentage) and their impact on energy use intensity (EUI) and outdoor thermal comfort (OTC) in the hot-summer Mediterranean city of Irbid, Jordan. Detailed numerical and physical analyses have been performed, including the effect of different climatic factors with different weather conditions (annual, summer and winter) and different spatial zones.

By applying the study approach, it was concluded that there is a negative correlation between the canyon aspect ratio and the urban block design performance. The canyon aspect ratio is an important parameter that is affected by other urban block parameters, including building height, the distance between buildings, setbacks or street width, as well as built-up percentage. Therefore, this study recommends that urban designers and decision-makers take these into account during the early design stages.

The study results showed the importance of the numerical and physical investigations in detailed spatiotemporal scales, pointing out the following:

- Controlling urban block design performance during the summer season in hot-summer Mediterranean climates is more valuable.
- North–south street canyons are more effective in enhancing microclimates; decreasing the Av. UTCI within north–south street canyons is more effective in enhancing the outdoor thermal comfort for the entire urban block design; therefore, increasing the length of the north–south street canyon, which can be increased by increasing the facade facing east to west, enhances the outdoor thermal comfort, and vice versa for the east–west street canyons.
- In the summer, the most effective weather factor for outdoor thermal comfort is solar radiation, while in the winter it is wind speed. During the summertime, the solar radiation effect overpowers the wind speed effect on outdoor thermal comfort, whereas in the winter, the wind speed decreases the air temperature and increases the cold sensation.
- The canyon aspect ratio should not be less than 1.5 and more than 4, where it will be fully exposed to solar radiation in the case of 1.5 and will be negatively affected in the case of more than 4 by the longwave radiation during the summer season. Furthermore, during the winter, increasing the canyon aspect ratio will slow the wind speed, which decreases the cold sensation.
- The setbacks should be studied side by side with the orientation. In our study area, the regulations mainly named the setbacks as side setbacks, front setbacks and back setbacks, where front setbacks could be facing the north, south, east or west, and since setbacks are considered half of the street width, this is not ideal, based on the canyon aspect ratio results.

In terms of energy use intensity, increasing building heights and canyon aspect ratios and decreasing the distances between buildings enhances the energy use performance due to reducing the solar radiation amount received on building walls during the summer, and since the summer season is the longest period in the study area, the cooling energy use will be reduced as well as the daylight potential.

Buildings with a north–south orientation have a higher energy use performance. This is emphasized by the fact that there is a correlation between outdoor and indoor environmental quality.

This research can help optimize urban block design performance during the early design stages as well as provide a numerical and visual analysis to understand the effect of different urban block designs on a microclimate for designers and decision-makers. Nonetheless, this is helpful for reducing the number of correlated parameters by including the desired range of effectiveness of these parameters on outdoor thermal comfort and energy use intensity. Thus, future work will focus on optimizing urban building forms using evolutionary algorithms based on outdoor thermal comfort since we reduced the ranges of the street canyon ratio (building heights and street widths) and orientation.

**Author Contributions:** Conceptualization, M.M.K.; Methodology, M.M.K.; Software, M.M.K.; Validation, M.M.K.; Formal analysis, M.M.K.; Investigation, M.M.K.; Resources, M.M.K.; Data curation, M.M.K.; Writing—original draft, M.M.K.; Writing—review & editing, M.M.K. and P.V.G.; Visualization, M.M.K.; Supervision, M.M.K. and P.V.G.; Project administration, M.M.K.; Funding acquisition, M.M.K. and P.V.G. All authors have read and agreed to the published version of the manuscript.

**Funding:** This research received no external funding.

**Institutional Review Board Statement:** Not applicable.

**Informed Consent Statement:** Not applicable.

**Data Availability Statement:** Not applicable.

**Acknowledgments:** The authors would like to express their sincere thanks to everyone who contributed to this research.

**Conflicts of Interest:** The authors declare no conflict of interest.

## Abbreviations

| Abbreviation | Definition |
| --- | --- |
| OTC | Outdoor thermal comfort |
| CAR | Canyon aspect ratio |
| SVF | Sky-view factor |
| UTCI | Universal thermal climate index |
| BUP | Built-up percentage |
| TMY | Typical meteorological year |
| CFD | Computational fluid dynamics |
| $Z_0$ | Roughness length |
| HB | Honeybee component |
| CSP | Cold sensation percentage |
| CAD | Computer-aided design |
| SST | Shear–stress transport |
| $U_{ref}$ | Wind speed at height $Z_{ref}$ |
| EUI | Energy use intensity |
| UHI | Urban heat island |
| MRT | Mean radiant temperature |
| SRI | Solar reflective index |
| UWG | Urban weather generator |
| UBL | Urban boundary layer |
| RANS | Reynolds-averaged Navier–Stokes |
| DF | Dragonfly component |
| TCP | Thermal comfort percentage |
| HSP | Heat sensation percentage |
| OpenFOAM | Open-source field operation and manipulation |
| $Z_{ref}$ | Height of the meteorological station |
| $Z_{Ground}$ | Ground level |

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
