# Peer review of "Outdoor Thermal Comfort Integrated with Energy Consumption for Urban Block Design Optimization: A Study of the Hot-Summer Mediterranean City of Irbid, Jordan"

_sustainability, doi:10.3390/su15108412_

Round 1

Reviewer 1 Report

In this manuscript, the authors tried to find out the relationships between urban morphology and thermal performance by using a comparative study.  The theme is very interesting and worthy reading. Overall, the manuscript was in a good structure. Especially, the authors used several simulation tools to conduct a good job. However, the following issues should be clarified!

The major concerns:

(1) The authors should provide a good scientific question although they did a lot of job.

(2) The abstract should be focused on new findings. Compared to the body of the manuscript, the abstract was not good enough.

(3) How to verify the simulation results?

(4) The manuscript is a little wordy.

The minor ones:

(1) What is CAR in the abstract?

(2) It is better to reduce the keywords.

(3) In line 136, it should be Over.

(4) In line 194, there is no need to use 31.963158. 31.96 is enough!

(5) In line 208, Selection?

It is strongly recommended to proofread the paper thoroughly again because it has many grammar problems.

It is strongly recommended to proofread the paper thoroughly again because it has many grammar problems.

Author Response

The authors took the whole comments carefully and please check the attached file.

Thank you for your comments, we appreciate it.

Reviewer 2 Report

Dear Authors

It is an interesting topic, however in some time I observe too indicators. As your mentioned in the objective "...correlations between different urban block design parameters and their environmental performance presented by energy use intensity (EUI) and outdoor thermal comfort (OTC),,," but the structure and data show it is not clear that. Suggestion, you can get better order in your presentation and put emphasys in the correlation ....

On the other hand, authors mentioned the indicators and data used, but not show data curation, statistical tratment, control etc, It is basic in order to follow the methodology.

Insist, I can not see the figures, but I hope the results are showed in ordwr and well tuning

The conclusions follow being results

Other comments are in the manuscript

Author Response

Point 1:  It is an interesting topic, however in some time I observe too indicators. As your mentioned in the objective "...correlations between different urban block design parameters and their environmental performance presented by energy use intensity (EUI) and outdoor thermal comfort (OTC),,," but the structure and data show it is not clear that. Suggestion, you can get better order in your presentation and put emphasys in the correlation ....

On the other hand, authors mentioned the indicators and data used, but not show data curation, statistical tratment, control etc, It is basic in order to follow the methodology.

Response 1: The authors a little bit confused about this point and tried to reconstruct the study results in the revised manuscript, however, there are two reasons to show data in this way:

  • The regulations parameters: we tried to connect the effects of urban design parameters and their impact while these parameters are listed as mentioned in the regulations in the study.
  • The intensive correlations between the parameters themselves, e.g. canyon aspect ratio (CAR) combines buildings' height and distances between buildings, the distance between buildings is double the setbacks while the buildings' height is fixed according to the buildings' regulations as we mentioned and to study the buildings' height alone we need a huge data to be enough and to fix both issues we tried to figure out the effect based on CAR and we resulted that buildings' height better to be studied side by side with distances between buildings. In addition, to study the buildings' height effect we should fix the distance between buildings with different buildings' heights we did that as we mentioned in the study results and methods and it gave the same results as the CAR effect which is correct but we tried to be more precise.

Point 2: Insist, I can not see the figures, but I hope the results are showed in ordwr and well tuning

Response 2: The study has done 59 iterations of modeling and simulation for 59 different urban block designs and each one has 7 different data (Av. UTCI, Av. CSP, Av. TCP, Av. HSP, cooling demand, heating demand, and EUI) and we would like to add it but it is impossible to insert these data for all urban block designs in figures, therefore, the study tried to explain it in charts. Also, the study inserted figures for the proposals at the end of the results.

Point 3: The conclusions follow being results

Response 3: Conclusions section has been reformulated taking into account this comment and the authors agree with this comment

The authors took the whole comments carefully and please check the attached files.

The pdf file you have uploaded and the whole comments within it have been answered and replied please check this file attached below.

Reviewer 3 Report

This study highlights the increasing importance of considering environmental quality in urban design, beyond just energy performance. It focuses on analyzing the correlations between different urban block design parameters and their impact on energy use intensity (EUI) and outdoor thermal comfort (OTC) in a hot-summer Mediterranean climate in Jordan. The study adopts a performance-driven approach using simulation tools and evaluates nine different urban block orientations. The results suggest a positive correlation between the compactness of urban blocks and their environmental performance, with north-south street canyons found to be more effective in enhancing microclimates. The study emphasizes the need to understand the distribution of open spaces formed by buildings and to strike a balance between day and night, as well as summer and winter conditions in outdoor spaces.

Overall, the study is well-structured and written. However, there are still some comments that need to be addressed in order to further improve its quality. Suggestions for improvement include:

1) Reorganize the introduction into two sections: Introduction and Related Works, to provide a more concise and focused overview.

2) Clearly summarize the contributions of the paper as bullet points in the Introduction section, to highlight the unique aspects and significance of the study.

3) Provide a clear and organized structure for the study at the end of the Abstract.

4) Define all acronyms and technical terms used in the study to enhance clarity and comprehension.

5) Conclude the study with a concise summary of the main findings and their implications for future research or practical applications.

inor editing of English language required. Please proofread the paper carefully.

Author Response

The authors took the whole comments carefully and answered these comments in the attached file below.

Thank you for your comments, we appreciate it.

Reviewer 4 Report

This is an interesting paper presenting research on outdoor thermal comfort and energy performance in correlation with building orientation. The authors sufficiently describe the methodology and provide a plausible overview of the state of the knowledge. The research is well described and the conclusions are clear and robust. I have some minor comments in regard to the presentation of the research in the paper.

1.       Please avoid using abbreviations in the abstract and conclusions

2.       Please be sure to explain each abbreviation when introducing it for the first time (e.g. CAR), ideally provide the list of abbreviations at the beginning of the paper.

3.     Please improve also visual appearance of the tables.

Despite these minor remarks, the paper is publishable and I recommend it to be published in Sustainability after minor revisions.

Author Response

(The authors gave the same response as above.)

Round 2

Reviewer 3 Report

The authors have carefully addressed my comments, I have no further suggestions.

The language as signficantly improved compared to the previous version of the article.